# Unexpected Pathogen Diversity Detected in Australian Avifauna Highlights Potential Biosecurity Challenges

**DOI:** 10.3390/v15010143

**Published:** 2023-01-02

**Authors:** Vasilli Kasimov, Michelle Wille, Subir Sarker, Yalun Dong, Renfu Shao, Clancy Hall, Dominique Potvin, Gabriel Conroy, Ludovica Valenza, Amber Gillett, Peter Timms, Martina Jelocnik

**Affiliations:** 1School of Science, Technology and Engineering, University of the Sunshine Coast, Sippy Downs, QLD 4557, Australia; 2Centre for Bioinnovation, University of the Sunshine Coast, Sippy Downs, QLD 4557, Australia; 3Sydney Institute for Infectious Diseases, School of Life and Environmental Sciences and Medical Sciences, The University of Sydney, Sydney, NSW 2006, Australia; 4Department of Microbiology and Immunology, The Peter Doherty Institute for Infection and Immunity, The University of Melbourne, Melbourne, VIC 3010, Australia; 5Department of Microbiology, Anatomy, Physiology and Pharmacology, School of Agriculture, Biomedicine and Environment, La Trobe University, Melbourne, VIC 3086, Australia; 6Australia Zoo Wildlife Hospital, Beerwah, QLD 4519, Australia

**Keywords:** Australia, avipoxvirus, beak and feather disease virus, birds, biosecurity, chlamydia, *Columbid alphaherpesvirus 1*, herpesvirus, *Psittacid alphaherpesvirus 1*, wildlife disease

## Abstract

Birds may act as hosts for numerous pathogens, including members of the family *Chlamydiaceae*, *beak and feather disease virus* (BFDV), avipoxviruses, *Columbid alphaherpesvirus 1* (CoAHV1) and *Psittacid alphaherpesvirus 1* (PsAHV1), all of which are a significant biosecurity concern in Australia. While *Chlamydiaceae* and BFDV have previously been detected in Australian avian taxa, the prevalence and host range of avipoxviruses, CoAHV1 and PsAHV1 in Australian birds remain undetermined. To better understand the occurrence of these pathogens, we screened 486 wild birds (kingfisher, parrot, pigeon and raptor species) presented to two wildlife hospitals between May 2019 and December 2021. Utilising various qPCR assays, we detected PsAHV1 for the first time in wild Australian birds (37/486; 7.61%), in addition to BFDV (163/468; 33.54%), *Chlamydiaceae* (98/468; 20.16%), avipoxviruses (46/486; 9.47%) and CoAHV1 (43/486; 8.85%). Phylogenetic analysis revealed that BFDV sequences detected from birds in this study cluster within two predominant superclades, infecting both psittacine and non-psittacine species. However, BFDV disease manifestation was only observed in psittacine species. All *Avipoxvirus* sequences clustered together and were identical to other global reference strains. Similarly, PsAHV1 sequences from this study were detected from a series of novel hosts (apart from psittacine species) and identical to sequences detected from Brazilian psittacine species, raising significant biosecurity concerns, particularly for endangered parrot recovery programs. Overall, these results highlight the high pathogen diversity in wild Australian birds, the ecology of these pathogens in potential natural reservoirs, and the spillover potential of these pathogens into novel host species in which these agents cause disease.

## 1. Introduction

Humans, domestic animals, wildlife, and their shared environment serve a crucial relationship in infectious disease transmission and emergence, with most infectious diseases identified in humans being of other animal origins [1,2]. Avian species are hosts for many viral, protozoal and bacterial pathogens, some of which may pose significant biosecurity risks and provide a substantial opportunity for spillover infection to wildlife, livestock and humans [3]. Over the past decades, numerous Australian studies have discovered and isolated several novel avian pathogens, significantly increasing our knowledge of the avian microbiome and the potential spillover risks associated with these pathogens [4,5,6]. Despite these findings, we still know little about many of these pathogens’ genetic and host diversity and their rate of occurrence in Australian birds.

Most surveillance studies of Australian wild birds focus on zoonotic avian diseases such as avian influenza A virus and other infectious agents contributing to economic loss in poultry production, including Newcastle disease virus and Marek’s disease virus [7,8,9]. Some wildlife pathogens, such as *Beak and feather disease virus* (BFDV), are well-studied in Australia [10]. However, the prevalence of other pathogens that may contribute to the decline in avian wildlife populations, including herpesviruses (*Columbid alphaherpesvirus 1* (CoAHV1) and *Psittacid alphaherpesvirus 1* (PsAHV1)) and members of the avipoxviruses and *Chlamydia* genus remain understudied [5,6,11,12,13]. Australia and its offshore islands host over 890 avian species, with approximately 45% of them being endemic [14]. Additionally, many globally diverse avian species are hypothesised to have originated from Australia, including all members of the orders Columbiformes (pigeons and doves) and Psittaciformes (parrots) [15,16]. Pigeons and parrots are predated by and share their habitat with various predatory birds, such as raptors (falcons, hawks, kites and owls) and kingfishers [17,18]. Therefore, due to the potential interactions between these species, it is important to understand the host range, ecological impacts, distribution and potential risks of spillover of wildlife pathogens within Australian birds.

This study investigated the prevalence and detection of BFDV, CoAHV1, PsAHV1, and members of the *Avipoxvirus* genus and *Chlamydiaceae* family, all being infectious agents of concern in Australian wild bird populations. We utilised samples from 486 wild birds, including kingfishers, parrots, pigeons and raptors that were collected from various locations across Southeast Queensland and admitted to two wildlife hospitals from Southeast Queensland, Australia. Samples were screened using a combination of family, genus and species-specific qPCR assays, followed by preliminary molecular characterisation of infecting strains utilising conventional PCR. Utilising this approach, we detected these pathogens within a broad range of traditional and novel avian hosts, further expanding our understanding of the current host range, genetic diversity and role in the coinfection of these pathogens in Australian avifauna.

## 2. Materials and Methods

### 2.1. Ethics Statement

All samples were sourced from euthanised birds collected across Southeast Queensland that had been admitted for various reasons by the public to the Australia Zoo Wildlife Hospital (AZWH, Beerwah, QLD, Australia) and The Royal Society for the Prevention of Cruelty to Animals (RSPCA, Brisbane, QLD, Australia). Pre-sampling, admission, care, and euthanasia were all conducted by attending veterinarians. Sample and swab collection from the euthanised birds was approved by the University of the Sunshine Coast Animal Research Ethics Committee (ANE1940, ANE2057, ANS1860).

### 2.2. Sample Collection

This study utilised 827 samples from a total of 486 birds (where in some instances, two samples were obtained from the same bird), encompassing 10 avian families and 44 species admitted to the Australia Zoo Wildlife Hospital (AZWH, Beerwah, QLD, Australia) and RSPCA (RSPCA, Brisbane, QLD, Australia) between May 2019 and December 2021. Each bird was then categorised based on their taxonomic groups, including kingfishers (Coraciiformes: *n* = 86), parrots (Psittaciformes: *n* = 262), pigeons (Columbiformes: *n* = 97) and raptors (Accipitriformes, Falconiformes and Strigiformes: *n* = 41) (Figure 1A). Of these, 627 samples (627/827) were from 317 birds (317/486) described in our previous study, evaluating the prevalence and genetic diversity of *Chlamydiaceae* and their role in coinfection with BFDV [11]. The remaining 200 (200/827) samples were collected from 169 (169/486) new birds in the present study. The final samples used in this study consisted of pooled ocular/choanal (*n* = 345), cloacal (*n* = 340), pooled ocular/choanal/cloaca (*n* = 3) dry swabs, in addition to liver (*n* = 138) and lung (*n* = 1) tissue samples. All dry swabs were obtained from euthanised birds admitted to the AZWH and the Brisbane RSPCA. Various metadata, including the date, admission cause, location, and clinical manifestations, were recorded for each sampled bird (Appendix A).

### 2.3. Sampled Avian Species and Hospital Admission Causes

Of the four sampled avian groups targeted in this study, parrots (262/486; 53.9%) were the most commonly sampled groups, followed by pigeons (97/486; 20%), kingfishers (86/486; 17.7%) and raptors (41/486; 8.4%) (Figure 1A). All sampled birds were admitted to the animal hospitals under three leading causes: trauma (262/486; 53.91%), clinical disease (121/486; 24.90%) and animal attack (35/486; 7.20%). Vehicle collision (HBC) accounted for the most considerable amount of specified hospital admissions (60/486; 12.35%), followed by the clinical manifestation of psittacine beak and feather disease virus (PBFD) (49/486; 10.08%) and physical trauma resulting in bone fractures (47/486; 9.88%) (Figure 1B). Birds were also admitted due to other causes (18/486; 3.70%), such as the inability to fly, pulmonary contusion, land clearing, neurological and other unspecified issues (50/486; 10.29%) (Appendix A).

### 2.4. New Sample Processing and DNA Extraction

The 200 new samples, consisting of 62 mucosal swabs and 138 liver (5 mm × 5 mm) tissue samples, were processed in a Biosafety Cabinet by vortexing and heat lysis at 90 °C for 10 min, followed by DNA extraction using the QiaAMP DNA mini kit according to the manufacturer instructions (Qiagen, Australia). Following DNA extraction, all samples were stored in a −20 °C freezer until further analyses.

### 2.5. Chlamydiaceace, BFDV, Avipoxvirus, CoAHV1 and PsAHV1 qPCR Detection

The *Chlamydiaceae* and BFDV DNA detection was performed on all 200 new samples using the *Chlamydiaceae* family-specific probe-based qPCR targeting the 110 bp fragment of the chlamydial 23S ribosomal RNA gene [19]. For BFDV, we used a qPCR assay targeting a 495 bp fragment of the ORF C1 capsid protein [6] (Appendix A). The assays were performed as previously described by Kasimov et al. [11]. For the 627 previous samples, we used the *Chlamydiaceae* and BFDV qPCR results as acquired in our previous study in conjunction with our newly acquired results to recalculate the overall prevalence of these pathogens [11].

To confirm the presence of CoAHV1 and PsAHV1 DNA, all extracted DNA samples (*n* = 827) were subsequently screened with species-specific Sybr Green-based qPCR assays targeting 208 bp and 281 bp of the CoAHV1 UL30 and PsAHV1 UL16/17 gene, respectively [20,21]. All extracted DNA samples (*n* = 827) were also screened for avipoxviruses utilising a qPCR assay amplifying a 578 bp segment of the *Avipoxvirus* P4b gene [22] (Appendix A). 

The CoAHV1, PsAHV1 and *Avipoxvirus* qPCR assays were carried out in a total volume of 15 µL, consisting of 7.5 µL iTaq Universal SYBR Green Supermix (Bio-Rad, South Granville, Australia), 3.5 µL PCR grade water, 0.5 µL of each 10 µM forward and reverse primer and 3µL DNA template. All samples were run in duplicate, and positive (targeted organism gDNA or gene fragment (Synthetic PsAHV1 DNA; Genscript, Piscataway, United States)) and negative (MilliQ H_2_O) controls were included in each assay. The qPCR conditions were as follows: 95 °C for 3 min; 35 cycles of 95 °C for 15 s, with annealing temperatures being 55 °C for *Avipoxvirus* and PsAHV1, and 62 °C for CoAHV1 for 25 s, 72 °C for 30 s, and a final extension on 72 °C for 7 min. Birds were considered positive for screened pathogens if the targeted viral DNA was detected in duplicate from a single anatomical site, had a Cq value ≤ 33 and high-resolution melts (HRMs) of 81.0 °C, 84.5 °C, and 77.5 °C +/− 0.5 °C, for CoAHV1, PsAHV1 and *Avipoxvirus*, respectively (Appendix A). Therefore, a bird was considered positive for screened pathogens if a sample from a single or multiple anatomical sites were positive. For all qPCR assays in this study, samples with discordant results (those with only one replicate amplifying) or suspected inhibited amplification were retested.

### 2.6. Molecular Characterisation of Avipoxvirus, BFDV and PsAHV1

In order to provide an initial molecular characterisation of the detected strains, we amplified 717 bp fragments of the BFDV ORF V1 gene [23], 578 bp fragments of the *Avipoxvirus P4b* gene [22,24] and 667bp fragments of the PsAHV1 UL16/17 gene (Tomaszewski et al., 2003) using all BFDV, *Avipoxvirus* and PsAHV1 positive samples initially detected by the qPCR assays (Appendix A). The conventional PCR reactions were performed in 35 μL volume, consisting of 17.5 μL Amplitaq Gold mix (ThermoFisher, Australia), 12.5 μL PCR grade water, 1 μL of each 10 μM forward and reverse primers and 3 μL DNA template. The cyclingF conditions were as follows: initial denaturation at 95 °C for 10 min, followed by 35 cycles of 95 °C for 20 s, 57.0 °C for BFDV, 55.0 °C for *Avipoxvirus* and 59.5 °C for PsAHV1, followed by 72 °C for 45 s and a final extension at 72 °C for 7 min. Positive (targeted organism gDNA or gene fragment (Synthetic PsAHV1 DNA; Genscript, Piscataway, United States)) and negative (MilliQ water) controls were included in each assay. PCR products were electrophoresed on a 1.5% agarose gel, followed by visual confirmation under an ultraviolet (UV) transilluminator. Based on band intensity and DNA concentration, amplicons were bidirectionally Sanger sequenced at Macrogen (Macrogen, Seoul, South Korea). A total of 92 BFDV, four *Avipoxvirus* and 14 PsAHV1 amplicons were chosen for sequencing.

The amplification of CoAHV1 fragments was attempted using primers previously described by Phalen et al., targeting the UL30 gene [25]. However, due to poor DNA quality and/or low copy numbers of CoAHV1 DNA, the Sanger sequencing reads were poor quality and unresolved. Additionally, due to all of the newly acquired avian samples that were *Chlamydiaceae*-positive being of low chlamydial load (Ct values > 35) and samples from our previous study that were already thoroughly investigated for genetic diversity of *Chlamydiaceae* [11], further molecular characterisation of *Chlamydiaceae*-positive samples was not performed.

### 2.7. Sequence and Phylogenetic Analysis

Chromatogram sequence quality and analyses were performed using Geneious Prime 2022.1.1 (licensed software available from https://www.geneious.com (accessed on 29 October 2022)), according to the criteria that both resulting chromatograms for each sequenced amplicon should be of high quality (Phred quality score ≥ 30) and covered ≥ 90% of the amplified sequence length. We successfully resolved 89 BFDV, 14 PsAHV1 and four *Avipoxvirus* fragments. The remaining sequences did not meet the specified criteria, potentially due to failed sequencing and/or low DNA concentration. Sequence identity was confirmed using BLASTn (Altschul et al., 1990) and the nr/nt database (Appendix A). The acquired sequences from this study were deposited in GenBank under the following accession numbers: OP131423–OP131511 (BFDV), OP13512–OP131515 (*Avipoxvirus*), OP146304–OP14317 (PsAHV1).

Sequences generated in this study were aligned using the MAFFT algorithm (implemented in Geneious Prime) with all publicly available reference sequences in GenBank. Final alignment lengths were 631 nt for BFDV, 622 nt for PsAHV1, and 559 nt for *Avipoxvirus* sequences. Maximum likelihood phylogenetic trees were then constructed for each alignment using IQ-TREE 2 [26], utilising 10,000 bootstrap replicates. IQTREE2 automatically incorporated the best fit model of nucleotide substitution for each sequence alignment: HKY+F+I+G4 for *Avipoxvirus*, TIM3+F+I+G4 for BFDV and K2P for PsAHV1. Both the *Avipoxvirus* and BFDV ML trees were midpoint rooted, whilst Cacatuid alphaherpesvirus 2 (MK360902) was utilised as an outgroup for the PsAHV1 ML tree.

### 2.8. Statistical Analysis

Sample size calculations to determine apparent prevalence within a 95% confidence interval (CI) were performed using the Epitools sample size calculation tool (epitools.ausvet.com.au/oneproportion) [27]. To estimate the apparent prevalence of pathogens screened in this study within a 95% confidence interval (CI) and a precision of ± 5%, we assumed an *Avipoxvirus*, BFDV, *Chlamydiaceae*, CoAHV1 and PsAHV1 prevalence of 15%, 35%, 10%, 10% and 1%, respectively. The estimated true and apparent prevalence of avipoxviruses, BFDV, *Chlamydiaceae*, CoAHV1 and PsAHV1 infections from testing results using a test of known sensitivity (0.90) and specificity (0.95) within a 95% CI was determined using the estimating prevalence utility tool (epitools.ausvet.com.au/trueprevalence), as implemented in Epitools. IBM SPSS Statistics software (IBM, Sydney, NSW, Australia) was utilized to perform chi-squared analysis, binomial logistic regression, estimate odds ratios, and determine the presence, direction, and magnitude of potential correlations between coinfection for each pathogen and various recorded metadata. *p*-values ≤ 0.05 were considered statistically significant.

## 3. Results

### 3.1. Beak and Feather Disease Virus Prevalence

BFDV DNA was detected in one-third of birds (33.54%; 163/486) and within all sampled avian orders (Table 1). Unexpectedly, pigeons rather than parrots had the highest prevalence of BFDV at 40.21% (39/97) (Figure 2), with BFDV DNA commonly detected in crested pigeons (*Ocyphaps lophotes*) (19/39; 48.72%). Pigeons only infected with BFDV and displayed signs of disease included two crested pigeons (2/19; 10.53%), diagnosed as clinically unwell by the attending veterinarians (Appendix A).

Parrots had the second-highest prevalence of BFDV at 35.50% (93/262) (Figure 2). Between the two sampled parrot families, *Cacatuidae* had a significantly higher proportion (*p*-value < 0.0001) of individuals detected with BFDV (43/71; 60.56%) compared to *Psittaculidae* species (50/191; 26.18%) and were 4.4 times more likely to show signs of PBFD (*p*-value = 0.001; exp(B) = 4.353) (Appendix A). Within each parrot family, BFDV was the most prevalent in sulphur-crested cockatoos (*Cacatua galerita*, family *Cacatuidae*) (18/24; 75.00%) and scaly-breasted lorikeets (*Trichoglossus chlorolepidotus*, family *Psittaculidae*) (18/42; 42.86%) (Appendix A). Of birds infected with BFDV only, 18 parrots, all of which were in the genus *Trichoglossus* (lorikeets), were only infected with BFDV and showed signs of disease, including psittacine beak and feather disease (PBFD) (12/18; 66.67%), emaciation (3/18; 16.67%), lorikeet paralysis syndrome (LPS) (2/11; 11.11%), and individuals considered clinically unwell by attending veterinarians (1/18; 5.56%) (Appendix A). Although BFDV was detected in the majority of species in this study (31/45; 68.89%), PBFD was only evident in six species of parrots: the galah (*Eolophus roseicapilla*), little corella (*Cacatua sanguinea*), long-billed corella (*Cacatua tenuirostris*), rainbow lorikeet (*Trichoglossus moluccanus*), scaly-breasted lorikeet, and sulphur-crested cockatoo (all of which tested positive for BFDV) (Appendix A).

BFDV DNA was detected from all sampled anatomical sites but was 1.7 times more likely to be found in liver samples than in eye/choana and cloacal swabs (exp(B) = 1.656; *p*-value = 0.008). Unsurprisingly, birds presenting with PBFD were 9.2 times more likely to be detected with BFDV (exp(B) = 9.195; *p*-value < 0.001). Lastly, BFDV DNA was 1.7 times more likely to be detected in pigeons than in kingfishers (exp(B) = 1.703; *p*-value = 0.024). No other significant relationships exist between BFDV detection and clinical disease or sampled avian categories in this study.

**Figure 2 viruses-15-00143-f002:**
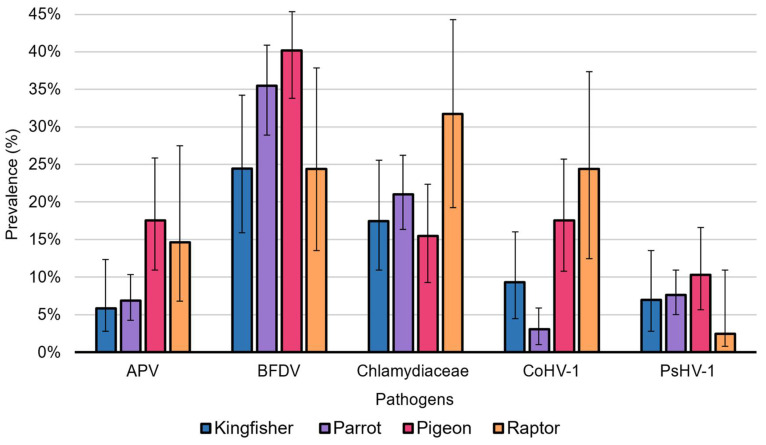
Pathogen prevalence within each sampled avian category. This histogram compares the percentage prevalence (*y*-axis) of *Avipoxvirus*, BFDV, *Chlamydiaceae*, CoAHV1 and PsAHV1 (*x*-axis) between each sampled avian category in this study. Histogram error bars represent the 95% confidence interval for the overall prevalence of each pathogen detected within sampled avian categories.

**Table 1 viruses-15-00143-t001:** qPCR detection rates of *beak and feather disease virus* (BFDV), *Chlamydiaceae*, *psittacid alphaherpesvirus 1* (PsAHV1), *columbid alphaherpesvirus 1* (CoAHV1) and *Avipoxvirus* in birds and swab samples from this study.

	Total Pos.	Apparent Prevalence (%) ^a^	True Prevalence (%) ^b^
Birds BFDV pos.	163/486	33.54 (CI 29.49–37.85)	33.58 (CI 28.81–38.65)
Swabs BFDV pos.	212/688	30.81 (CI 27.48–34.36)	30.37 (CI 26.45–34.54)
Tissue BFDV pos.	59/139	42.45 (CI 34.54–50.76)	44.05 (34.75–53.83)
Birds *Chlamydiaceae* pos.	98/486	20.16 (CI 16.84–23.96)	17.84 (CI 13.93–22.31)
Swabs *Chlamydiaceae* pos.	138/688	20.06 (CI 17.24–23.21)	17.72 (CI 14.40–21.43)
Tissue *Chlamydiaceae* pos.	1/139	0.72 (CI 0.13–3.96)	NA
Birds PsAHV1 pos.	37/486	7.61 (CI 5.57–10.32)	3.07 (CI 0.67–6.26)
Swabs PsAHV1 pos.	23/688	3.34 (CI 2.24–4.97)	NA
Tissue PsAHV1 pos.	20/139	14.39 (CI 9.51–21.18)	11.05 (CI 5.31–19.04)
Birds CoAHV1 pos.	43/486	8.85 (CI 6.63–11.71)	4.53 (CI 1.92–7.89)
Swabs CoAHV1 pos.	40/688	5.81 (CI 4.30–7.82)	0.96 (CI 0.00–3.32)
Tissue CoAHV1 pos.	9/139	6.47 (CI 3.44–11.85)	1.74 (CI 0.00–8.06)
Birds *Avipoxvirus* pos.	46/486	9.47 (CI 7.17–12.39)	5.25 (CI 2.55–8.70)
Swabs *Avipoxvirus* pos.	36/688	5.23 (CI 3.80–7.16)	0.27 (CI 0.00–2.54)
Tissue *Avipoxvirus* pos.	17/139	12.23 (CI 7.78–18.71)	8.51 (CI 3.27–16.13)

NA: Unable to measure true prevalence based on the sample size and the number of positive samples; Swabs: Samples include pooled eye/choana and cloacal swabs; Tissue: Samples include liver and lung samples. a Wilson CI; b Blaker CI testing results using a test of known test sensitivity 90%, test specificity 95%.

### 3.2. Chlamydiaceae Prevalence

*Chlamydiaceae* was detected in one-fifth of all sampled birds (98/486; 20.16%) (Table 1) and was most prevalent in raptor species (13/41; 31.71%) (Figure 2), particularly in the order Accipitriformes (7/17; 41.18%). However, only one raptor, a collared sparrowhawk (*Accipiter cirrocephalus*) which was infected only with *Chlamydiaceae,* presented with signs of morbidity (1/13; 15.38%) (Appendix A). The second-highest detection rate of *Chlamydiaceae* was found in parrots at 20.99% (55/262) (Figure 2). A significant disparity in chlamydial prevalence was noteworthy between the two sampled parrot families: *Cacatuidae* (24/71; 33.80%) had more than double the chlamydial prevalence and were 2.6 times more likely to be detected with *Chlamydiaceae* (exp(B) = 2.636; *p*-value = 0.002), compared to *Psittaculidae* (31/191; 16.23%) (Appendix A). Additionally, within the two sampled parrot families, *Chlamydiaceae* prevalence was the highest in the galah (family *Cacatuidae*) (10/23; 43.48%) and Australian king-parrot (*Alisterus scapularis*, family *Psittaculidae*) (5/10; 50.00%). 

Seven individuals were infected with *Chlamydiaceae* only and showed clinical signs of disease, including an emaciated scaly-breasted lorikeet and yellow-tailed black cockatoo (*Zanda funerea*), a galah and rainbow lorikeet with feather loss, two rainbow lorikeets with LPS, and a pale-headed rosella (*Platycercus adscitus*) that was deemed clinically unwell (Appendix A). Chlamydial DNA was almost exclusively detected in eye/choana and cloacal samples (138/688; 20.06%) and was significantly more likely to be detected in these sites compared to liver and lung tissue samples (1/138; 0.72%) (*p*-value < 0.001). The only instance of chlamydial DNA detection from a tissue sample was from a liver isolated from a single little corella diagnosed with chlamydiosis, severe conjunctivitis and emaciation, which was also coinfected with BFDV. There was no significant relationship between *Chlamydiaceae* detection and clinical disease in this study (*p*-value > 0.05).

### 3.3. CoAHV1 Prevalence

CoAHV1 was detected for the first time within a series of novel Australian avian hosts, including various kingfisher, parrot, pigeon, and raptor species, with a total prevalence of 8.85% (43/486) (Table 1). CoAHV1 prevalence was the highest in raptors (10/41; 24.39%) (Figure 2), particularly within barn owls (*Tyto alba*) (5/10; 50.00%). However, there were no cases of raptor species detected with CoAHV1 only and presenting with any clinical disease (Appendix A). Pigeons had the second-highest prevalence of CoAHV1 (17/97; 17.53%) (Figure 2), with CoAHV1 almost exclusively detected in crested pigeons (12/17; 70.59%) (Appendix A). Like raptor species, all pigeons detected with CoAHV1 only appeared asymptomatic. The only case of a bird that was infected with CoAHV1 only and showed signs of disease in this study was from a single sacred kingfisher (*Todiramphus sanctus*) diagnosed with emaciation and cataracts (Appendix A). Of all sampled avian categories, raptors were significantly more likely to be detected with CoAHV1 when compared to other avian categories, except pigeons (*p*-value < 0.03). In contrast, parrots were the least likely and had the lowest CoAHV1 prevalence of 3.05% (8/262; *p*-value < 0.02). This study showed no statistical difference between CoAHV1 detection and clinical disease or sample type (*p*-value > 0.05).

### 3.4. PsAHV1 Prevalence

PsAHV1 had the lowest overall prevalence of 7.61% (37/486) (Table 1) among all sampled pathogens. This study detected, for the first time, PsAHV1 in wild Australian parrots and within previously undescribed hosts, including kingfisher, raptor, and pigeon species. Pigeons had the highest detection rate of PsAHV1 (10/97; 10.31%) (Figure 2), with all infected individuals being crested pigeons (10/10; 100.00%) (Appendix A). Parrots had the second-highest detection rate of PsAHV1 at 7.63% (20/262) (Figure 2), with the vast majority of infected birds being rainbow lorikeets (13/20; 65.00%) (Appendix A). There were no cases of birds infected with PsAHV1 only and presenting with disease (Appendix A). PsAHV1 was more than four times more likely to be detected in the liver compared to eye/choana and cloacal samples (exp(B) = 4.859; *p*-value < 0.001). However, no significant relationship exists between PsAHV1 infection and disease or between avian categories in this study (*p*-value > 0.05).

### 3.5. Avipoxvirus Prevalence

*Avipoxvirus* was detected in approximately one-tenth of all birds in this study (46/486; 9.47%) (Table 1) and all sampled avian categories. *Avipoxvirus* was most abundant in pigeon species (17/97; 17.53%) (Figure 2) and almost exclusively detected in crested pigeons (14/17; 82.35%) (Appendix A). Raptors had the second-highest prevalence of *Avipoxvirus* (6/41; 14.63%) (Figure 2), with half of the raptor positivity attributed to barn owls (3/6; 50.00%) (Appendix A). No cases of birds infected with *Avipoxvirus* only presented with clinical disease in this study (Appendix A). Upon comparing avian categories, pigeons were approximately three times more likely to be detected with *Avipoxvirus* DNA compared to kingfishers (exp(B) = 3.44; *p*-value = 0.02) and parrots (exp(B) = 2.88; *p*-value < 0.01). Furthermore, *Avipoxvirus* DNA was 2.5 times more likely to be detected from eye/choana and cloacal swabs than liver samples (exp(B) = 2.524; *p*-value = 0.003). No significant relationship exists between *Avipoxvirus* detection and clinical signs of disease in this study (*p*-value > 0.05).

### 3.6. Pathogen Coinfection and the Manifestation of Clinical Disease

Approximately one-quarter (120/486; 24.69%) of birds in this study presented with clinical disease. Therefore, we investigated the potential relationships between disease manifestation and pathogen coinfection. In total, 17.28% (84/486) of birds were coinfected with two or more pathogens (Figure 3), with 26.19% (22/86) of coinfected individuals presenting with clinical disease, including PBFD (13/22; 59.09%), emaciation (4/22; 18.18%), blindness (1/22; 4.55%), LPS (1/22; 4.55%) or were diagnosed as unwell (3/22; 13.64%) (Appendix A). Of all sampled birds, only a single sacred kingfisher (1/486; 0.21%) was coinfected with all five screened pathogens but remained asymptomatic with no evident signs of infection or clinical disease. 

Eight birds (8/486; 1.65%) were simultaneously coinfected with four pathogens, of which only two cockatoos presented with clinical disease, including an emaciated long-billed corella (coinfected with all four viruses) and a little corella presenting with severe feather loss (coinfected with all pathogens except BFDV). Twenty birds (20/486; 4.12%) were coinfected with three pathogens in various combinations, with one-quarter of them (5/20; 25.00%) presenting with symptoms, including two crested pigeons which were emaciated and moribund, respectively (both coinfected with *Avipoxvirus*, PsAHV1 and CoAHV1), a sulphur-crested cockatoo presenting with PBFD (coinfected with PsAHV1, BFDV and CoAHV1), and an emaciated little corella and barn owl (both coinfected with *Avipoxvirus*, BFDV, and CoAHV1) (Appendix A).

Fifty-five birds (55/486; 11.32%) were coinfected with two pathogens only, of which 29.09% (16/55) displayed clinical disease. The two most common pathogens birds were coinfected with were BFDV and *Chlamydiaceae* (29/55; 52.73%), with over half of coinfected individuals being parrots (16/29; 55.17%) (Appendix A). Over one-third of individuals coinfected with *Chlamydiaceae* and BFDV (11/29; 37.93%) displayed clinical symptoms such as PBFD (observed exclusively in parrots, 7/11; 63.64%), a clinically unwell crested pigeon and rainbow lorikeet (2/11; 18.18%), and a blind black kite (*Milvus migrans*) (1/11; 9.09%). The second most common pathogens birds were coinfected with were *Avipoxvirus* and PsAHV1 (11/55; 20.00%), with the vast majority of coinfected individuals being rainbow lorikeets (9/11; 81.82%). However, of all coinfected individuals, only three rainbow lorikeets presented with symptoms, including LPS, emaciation, and PBFD (3/11; 27.27%) (Appendix A). This study found no significant relationship between pathogen coinfection and clinical disease manifestation (*p*-value > 0.05).

**Figure 3 viruses-15-00143-f003:**
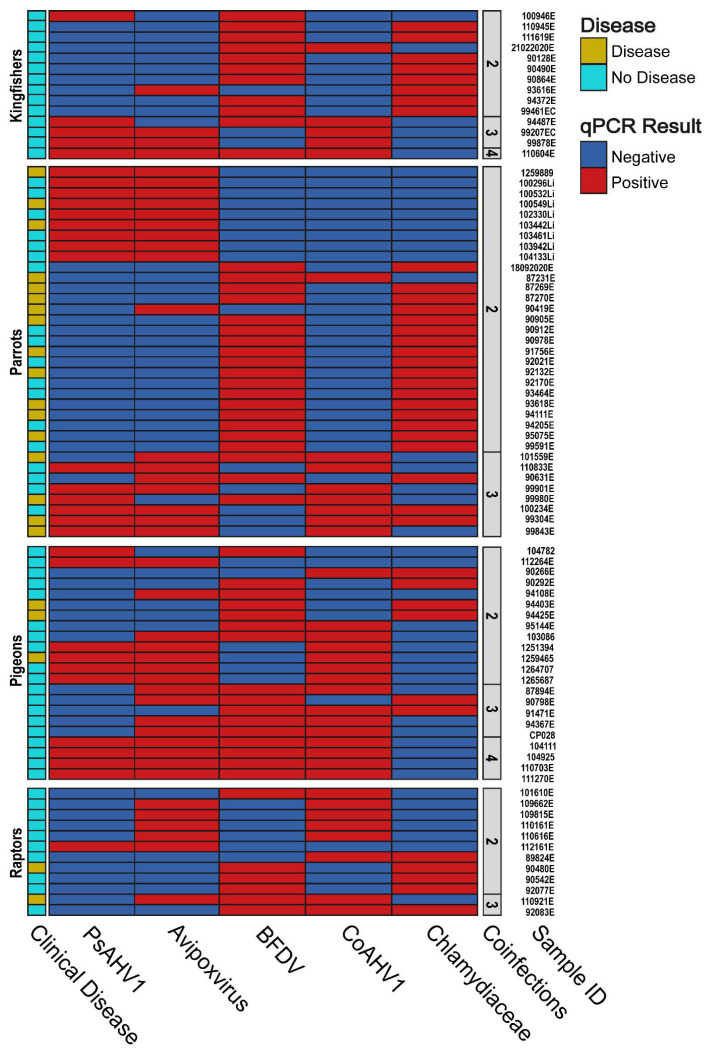
A heatmap showing the coinfections of two, three and four pathogens based on the qPCR results obtained from sampled birds in this study and whether clinical disease is present or absent. Birds are separated by avian category, with sequence identifications and number of coinfections provided along the *y*-axis.

### 3.7. Sequence Analysis of Avipoxvirus, BFDV and PsAHV1

#### 3.7.1. BFDV Phylogenetic Analyses

Phylogenetic analysis of BFDV sequences, comprising 89 partial BFDV *rep* gene sequences from this study and 145 global BFDV reference rep gene sequences, demonstrates the formation of two “superclades” (designated in this study as SC1 and SC2 for clarity only), driven by host variations (Figure 4A,B).

SC1 predominantly contains sequences detected from Australian *Psittaculidae* species, with 19/21 sequences detected from *Psittaculidae* species in this study (Australian king-parrots, rainbow lorikeets and scaly-breasted lorikeets) clustered into this clade, sharing a 97.98–99.90% similarity to sequences detected from Australian *Psittaculidae* species. An additional 12 sequences from this study, which clustered into this clade, were from three raptors (3/12), four pigeons (4/12) and five kingfishers (5/12), all sharing between 97.90–99.90% similarity to the same reference sequences detected from Australian lorikeets (Figure 4A).

SC2 contains more diverse global BFDV reference sequences detected from various psittacine hosts across different continents and is comprised of two clades (designated in this study as C1 and C2), with C2 having an additional four sub-clades (designated in this study as C2.1, C2.2, C2.3 and C2.4) (Figure 4A,B). Overall, SC2 appears more host-generalist, containing all 21 sequences detected from cockatoos in this study, two sequences from parrots, and 35 sequences from non-psittacine hosts. Out of the total of 58 sequences clustering within SC2, seven sequences from two parrots (2/7), three pigeons (3/7), and two kingfishers (2/7) formed a well-supported sub-clade within diverse sub-clade C2.1, sharing a 98.00–98.10% similarity to a BFDV sequence detected from a Rosy-faced lovebird (*Agapornis roseicollis*) from the UK.

Finally, an additional 14 sequences from six parrots (6/14), four pigeons (4/14), two kingfishers (2/14) and two raptors (2/14) clustered within a genetically diverse sub-clade C2.3, and 37 sequences from 15 parrots (15/37), 17 pigeons (17/37), three kingfishers (3/37) and two raptors (2/37) clustered within its own genetically diverse larger sub-clade C2.4, all sharing a 96.45–99.30% similarity with other Australian cockatoo BFDV reference sequences.

Clinical manifestation of BFDV infection (PBFD—indicated by a red circle next to sequence descriptions) was present in birds in both SC1 and SC2 but only manifested in psittacine species (24/89), despite identical sequences being detected from non-psittacine species. Within SC1, PBFD was evident from seven scaly-breasted lorikeets (7/9) and one rainbow lorikeet (1/8), whilst PBFD was apparent in two galahs (2/4), five little corellas (5/6), two long-billed corellas (2/2), and seven sulphur-crested cockatoos (7/9) within SC2.

**Figure 4 viruses-15-00143-f004:**
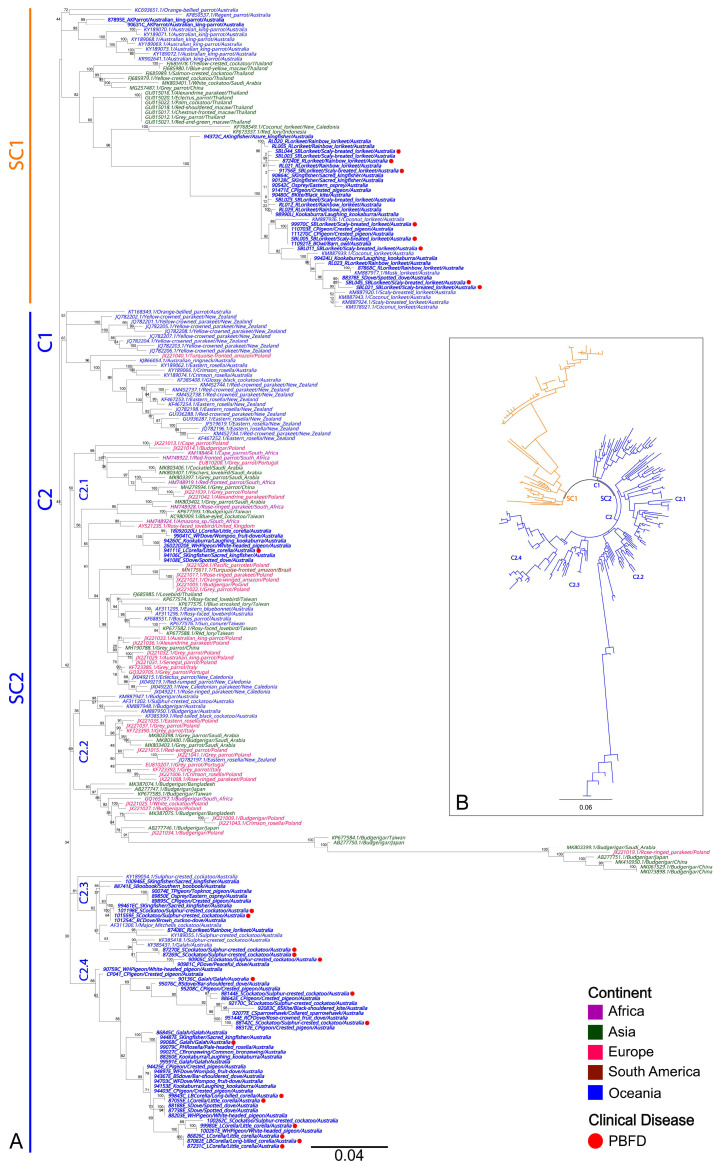
Phylogenetic relationships between 89 BFDV sequences detected from avian species in this study and 145 global reference strains (**A**). BFDV sequences are named by their corresponding GenBank accession number/host/country of origin and coloured text according to their continent of isolation. Sequences from Oceania are presented in blue, with sequences generated in this study being additionally boldened. A red circle is placed adjacent to sequences detected from birds that presented with disease signs consistent with PBFD. Bootstrap values are shown as a percentage. (**B**) The inset tree is broadly denoting BFDV SC1 and SC2 overall phylogeny between all publicly available BFDV *rep* gene sequences. The midpoint-rooted maximum likelihood phylogenetic trees were constructed using IQTREE2. The scale bar indicates the number of nucleotide substitutions per site.

#### 3.7.2. Avipoxvirus Phylogenetic Analyses

All four *Avipoxvirus* gene sequences from this study were 100.00% identical to each other; with the phylogenetic analysis of the *P4b* gene demonstrating that they all clustered within the *Avipoxvirus* sub-clade A3 (designated as per Gyuranecz et al.), which encompasses avipoxviruses previously described in an array of avian species including owls, pigeons and doves and seabirds (Figure 5) [28]. Within clade A3, all sequences generated in this study were further 100.00% similar to the *P4b* gene from an *Avipoxvirus* strain detected from a southern giant petrel (*Macronectes giganteus*) from Antarctica (KC017981) (Figure 5). Compounded with phylogenetic analysis, we used BLASTn to identify which viral species we detected, and the sequences generated here were 99.12% similar to the ICTV-ratified species, *Flamingopox virus* (NC_036582).

#### 3.7.3. PsAHV1 Phylogenetic Analyses

We generated 14 PsAHV1 UL16/17 gene sequences, which all shared 99.84–100.00% similarity to PsAHV1 sequences detected from various Brazilian psittacine species. Phylogenetic analyses show the formation of one main clade encompassing all 14 PsAHV1 UL16/17 gene sequences from this study and 18 PsAHV1 UL16/17 reference sequences detected from various psittacine species from Brazil (Figure 6). Three sequences detected from a sulphur-crested cockatoo, rainbow lorikeet and sacred kingfisher (99901C_SCockatoo, RL129_RLorikeet and 100946C_SKingfishker, respectively) share a 99.84% similarity to Brazilian reference sequences detected from a peach-fronted parakeet (*Eupsittula aurea*) and white-eyed parakeet (*Psittacara leucophthalmus*) (JF33861 and JF338872). The remaining 11 sequences from this study were 100.00% identical to all other Brazilian parrot reference strains, apart from three other reference sequences (JF338866, JF338867 and JF338871), which clustered together and formed their own sub-lineage. It is important to note that the conserved nature of the UL16/17 gene and the short fragment generated in this study (667bp) was not ideal for phylogenetic analysis. Therefore, we could not identify lineages with any certainty.

## 4. Discussion

### 4.1. Avipoxvirus, BFDV, Chlamydiaceae, CoAHV1 and PsAHV1 Detection in Wild Avian Hosts from Southeast Queensland

This study investigated five pathogens of biosecurity importance (avipoxviruses, BFDV, *Chlamydiaceae*, CoAHV1 and PsAHV1) within kingfisher, parrot, pigeon, and raptor species from Southeast Queensland admitted to wildlife hospitals. We detected all five pathogens (including the “exotic” PsAHV1) within all four sampled avian categories, albeit at varied detection rates (Figure 2). Coinfections with multiple pathogens were also common, although all were independent, and we did not detect significant patterns of coinfections (*p*-value > 0.05). Similar to our previous study, BFDV and *Chlamydiaceae* coinfection were the most prevalent and most commonly detected within psittacine species in this study [11].

This current study is the first to detect PsAHV1 in wild Australian birds (37/486; 7.61%) and within a series of novel hosts (kingfishers, raptors and pigeons). These results are significant, as PsAHV1 is considered an “exotic” pathogen in Australia and has only been detected once in Australia from two captive green-winged macaws (*Ara chloropterus*) imported from the United Kingdom in 2004 [29]. Since then, PsAHV1 has not been detected in Australia, most likely due to a lack of sampling and testing rather than the circulation of the virus. PsAHV1 is found in wild birds globally, with detection in birds from North and South America, Europe, the Middle East, Japan and New Zealand, and with disease manifestation reported in several captive Australian parrot species housed outside of Australia. Similar to this study, PsAHV1 has also been detected in a range of non-psittacine species suggesting this pathogen has a broad host range and is not limited to psittacine birds [29,30]. Apart from PsAHV1, novel alphaherpesviruses have been isolated from various psittacine species in Australia, including *cacatuid alphaherpesvirus 2* (CaHV-2) from a little corella, *psittacid alphaherpesvirus 3* (PsAHV-3) in eclectus parrots (*Eclectus roratus*), and *psittacid alphaherpesvirus 5* (PsAHV-5) from captive ringneck parrots (*Psittacula krameri*) [5,31,32].

CoAHV1 was the fourth most prevalent pathogen in this study (43/486; 8.85%), commonly detected in pigeon (17/43; 39.53%) and raptor species (10/43; 23.26%). These results are unsurprising, as CoAHV1 is a widespread and enzootic pathogen, particularly within rock dove (*Columba livia*) flocks in Melbourne and Sydney [25,33]. CoAHV1 has also been detected in a variety of Australian raptor species, including the powerful owl (*Ninox strenua*), Australian hobby (*Falco longipennis*) and barking owl (*Ninox connivens*), some of which were observed feeding upon feral pigeon carcasses infected with CoAHV1, likely resulting in their death [6,25,33,34]. Upon comparing detection results, the only other CoAHV1 surveillance study conducted in Australia was published by Phalen et al., detecting a notably higher prevalence of CoAHV1, ranging from 70.00 to 100.00% in Australian feral pigeon flocks [33]. The remaining Australian studies either opportunistically detected CoAHV1 via pan-herpesvirus PCR or were case reports of infected raptor species [6,25]. Our results show that CoAHV1 can circulate within a broader range of avian hosts. However, further studies are needed to assess whether the pathogenicity occurs in other avian species besides pigeons and raptors.

*Avipoxvirus* was the third most prevalent pathogen in this study (46/486; 9.47%) and was detected at significantly lower levels than BFDV and *Chlamydiaceae*. Multiple studies have genetically and molecularly characterised novel *Avipoxvirus* isolates in Australian avian species, including the Australian magpie (*Gymnorhina tibicen*), crimson rosella (*Platycercus elegans*), magpie-lark (*Grallina cyanoleuca*), shearwater species (*Procellariidae*) and silvereye (*Zosterops lateralis*) [13,35,36,37,38]. However, our study is the first to have conducted a large-scale surveillance study of this pathogen in wild Australian bird species. Despite the absence of Australian *Avipoxvirus* surveillance studies, a recent review demonstrated that *Avipoxvirus* prevalence could range up to 88% (in the case of an epizootic event), with a mean prevalence of 11.2% in islands vs. 2.3% in continents. Strikingly a 60% increase in the avian host range of avipoxviruses has been observed since a previous review published in 1999, with avipoxviruses now detected in over 374 wild bird species across 23 orders [39]. Therefore, the regular detection rate of novel avipoxviruses suggests that the full diversity and occurrence in wild birds remains undetermined, particularly within Australia.

Of the five screened pathogens, BFDV and *Chlamydiaceae* had the highest detection rates and were found in approximately one-third (163/483; 33.54%) and one-fifth (98/486; 20.16%) of sampled birds, respectively. BFDV detection rate in this study is consistent with other Australian reports, with recent Australian studies detecting between 31.00–38.10% within captive and wild bird populations, including psittacine and non-psittacine species [5,11,12,40,41]. The *Chlamydiaceae* prevalence of 20.16% in this study is similar to our previous study conducted in birds from Southeast Queensland (165/564; 29.26%) [11], likely due to 75.82% (627/827) of the same sample catalogue being used. Based on our previous study, at least 10% of *Chlamydiaceae* infections were attributed to the zoonotic *C. psittaci*. Interestingly, *Chlamydiaceae* were almost exclusively detected in mucosal swab samples and only detected from a single liver sample from a little corella with chlamydiosis (Appendix A). Other studies investigating *Chlamydiaceae* in avian liver samples also detected *Chlamydia* from 1/52 (1.90%) liver samples in pigeons and 0/243 (0.00%) of pooled organ tissue in migratory swifts over a nine-year study period [42,43]. These results potentially suggest that *Chlamydiaceae* infecting birds are usually asymptomatic infections and intermittently shed from the respiratory and gastrointestinal tract, and in rare cases of systemic infection, stress or immunosuppression are detected in high loads from organ tissue.

### 4.2. Pathogen Biosecurity Concerns and Precautions

Pathogens have enormous global impacts on animal populations, including wild birds, some of which are also migratory and are of significant biosecurity concern, as they are capable of importing and shedding diseases into our local environment [44]. For example, avian cholera (caused by the bacterium *Pasteurella multocida*) is driving the global decline of the Indian yellow-nosed albatross (*Thalassarche carteri*), with an 86.6% decrease from 1981 to 2016 [45]. Of current concern are the vast outbreaks of avian influenza, which have wiped out entire migratory seabird colonies since January 2022 [46]. Of the infectious agents included in this study, at least two (BFDV and PsAHV1) can have devastating impacts on parrot species [12,47]. There are over 370 parrot species in Australia, of which 20% are considered under threat, including 85 critically endangered and vulnerable species and 19 near risk of extinction [48]. Some of these critically endangered parrot species include the orange-bellied parrot (*Neophema chrysogaster*), swift parrot (*Lathamus discolor*) and various black-cockatoo species (*Zanda and Calyptorhynchus spp*) and the gang-gang cockatoo (*Callocephalon fimbriatum*) [49]. While none of these threatened taxa were included in our study, the fact that we have now demonstrated the prevalence of these pathogens in other Australian native parrot species is of concern. Although urbanisation, habitat loss and modification are the leading cause of population decline in psittacine species [50], infectious viral agents, such as BFDV and PsAHV1, are considered a significant threat to the persistence of robust populations of many psittacine species and add to the cumulative risk factors for threatened taxa. Avian species are additionally vectors for various zoonotic RNA viruses, which can cause disease within infected birds and humans, including influenza viruses, *West Nile virus* and *Usutu virus* [46,51]. However, the screening of these pathogens was beyond the scope of this study. 

BFDV in threatened parrot species is considered a key concern and was listed as a national threat in 2001 by the Australian Government under the Environment Protection and Biosecurity Conservation Act 1999 [52]. BFDV most likely originated and co-evolved within Australian parrots and has rapidly spread across the globe due to the exotic pet trade over the past 150 years and continues to threaten parrot conservation programs [6]. Species in which BFDV is commonly detected, including the little corella, pose a significant concern as they travel in large flocks, compete with other avian species for hollows and shed the virus into local environments, enabling the transmission and maintenance of BFDV endemicity [53]. Previously, a recombinant BFDV capsid protein vaccine was trialled on a limited number of psittacine species and was shown to induce an antibody response against the native BFDV. However, despite promising results, these studies are currently unfunded and are not underway [54]. Although no antiviral therapy is currently available to mitigate BFDV, effective management protocols implemented in zoological institutions, wildlife hospitals, veterinary clinics and rehabilitation centres, such as using potassium peroxymonosulphate between examinations, can inactivate BFDV and reduce viral spread and transmission [55]. 

PsAHV1 is the causative agent of Pacheco Disease and mucosal papillomas in parrot species, a highly infectious and lethal respiratory herpesvirus that usually causes the parrot to die within a few days of contracting the disease [56]. PsAHV1 is a highly lethal and infectious virus, with all psittacine species considered susceptible to infection [57]. Therefore, the detection of PsAHV1 for the first time in wild Australian birds raises significant biosecurity concerns regarding disease control and the conservation of threatened species. As previously mentioned, despite being considered an exotic pathogen, it is unsurprising that this virus is not circulating within wild bird populations [29]. However, it is interesting that previous Australian studies have not detected or attempted to screen for this virus prior to this study, as the unintended introduction of such an acute, highly infectious and lethal pathogen into endangered psittacine populations, such as the remaining population of the orange-bellied parrot, could result in a severe or total loss [6].

### 4.3. Phylogenetic Analyses of BFDV Sequences Shows the Formation of Lorikeet and Host-Generalist Clades

BFDV is a highly prevalent and widely disseminated pathogen with a host range of over 370 avian species. It is recognised for its high genetic diversity and flexible host-switching within the order Psittaciformes and even into distantly related non-psittacine species [58]. In the present study, we generated 89 BFDV rep gene fragments, of which 42 were detected from psittacine species and the remaining 47 from non-psittacine species. As previously stated, the phylogenetic tree construction of these and other sequences resulted in two distinct superclades (SC1 and SC2) (Figure 4), demonstrating the geographical clustering of sequences detected in Australian bird sequences. SC1 contains the majority of BFDV sequences detected from *Psittaculidae* species in this study, while SC2 contains all the cockatoo and the majority of non-psittacine BFDV sequences detected from birds in this study. These results are supported by a previous study by Das et al. [59], showing that BFDV sequences from lorikeet species showed strong tribe-specific clustering, forming distinct sub-clades monophyletic to sampling location and likely evolve independently. However, BFDV sequences detected from other hosts demonstrate a host generalist infectivity pattern and experience frequent inter-population admixture through inter-lineage recombination [59]. 

Lorikeet species compete closely with other native Australian avian species for nest hollows, which appear to be an important site of viral transmission [59]. This likely is why sequences identical to those detected from lorikeets were additionally detected from non-psittacine species in this study, including other hollow-dwelling species such as kookaburras and barn owls [60]. This hypothesis is additionally supported by previous studies that provide evidence for non-psittacine species, including kingfishers and owls, which have been shown to become infected with BFDV strains from lorikeet and other psittacine species if conditions are favourable for transmission [61,62]. Furthermore, predatory birds, including eagles, hawks, owls and kookaburras, also prey upon psittacine species and their young, which may serve as another important mode for BFDV transmission [17,63,64]. A limitation of our study is that we used only partial *rep* gene sequences, providing an initial phylogenetic context of BFDV diversity. Whole-genome sequencing and whole-genome derived phylogeny is needed to resolve BFDV diversity more accurately from our current study [59].

## 5. Conclusions

Our study provides novel findings and important data on the overall prevalence and coinfection of five significant pathogens of biosecurity importance (*Avipoxvirus*, *Chlamydiaceae*, BFDV, CoAHV1 and PsAHV1). We provided the first prevalence data of avipoxviruses from a large and diverse wild bird population from one region in Australia. Furthermore, we obtained PsAHV1 sequences from wild Australian birds for the first time, whilst expanding upon the currently known host range of BFDV and CoAHV1 in Australia. In doing so, we show that we do not yet clearly understand what is circulating in Australia due to the lack of testing on wild birds. Our findings highlight the need for further surveillance and molecular studies, particularly for PsAHV1 and BFDV, as the ecological impacts of these viruses could be devastating to recovery programs for threatened avian species. More data across host species and regions will allow for a better understanding of virus ecology and diseases of biosecurity concern.

## Figures and Tables

**Figure 1 viruses-15-00143-f001:**
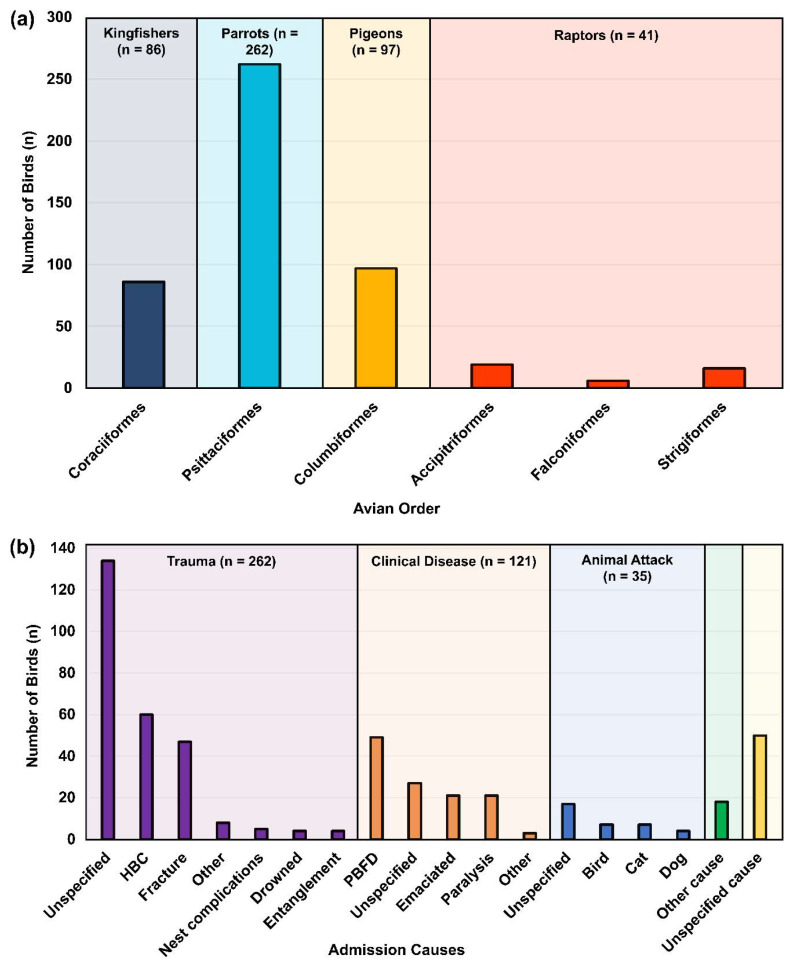
Number of sampled birds within each avian order and their admission causes. (**a**) Histogram displays the number of birds sampled (*y*-axis) within each avian order from this study (*x*-axis). (**b**) Histogram of the number of birds (*y*-axis) based on their admission cause (*x*-axis) to the AZWH and RSPCA Brisbane between May 2019 and December 2021. Within trauma, ‘Other’ includes eye injury, plane and window collision, neurotrauma and head trauma. Within clinical disease, ‘Other’ includes diarrhea, blindness, and poxvirus. ‘Other cause’ as an individual category includes the inability to fly, pulmonary contusions, land clearing, neurological issues, osteomyelitis, and severe soft tissue injuries.

**Figure 5 viruses-15-00143-f005:**
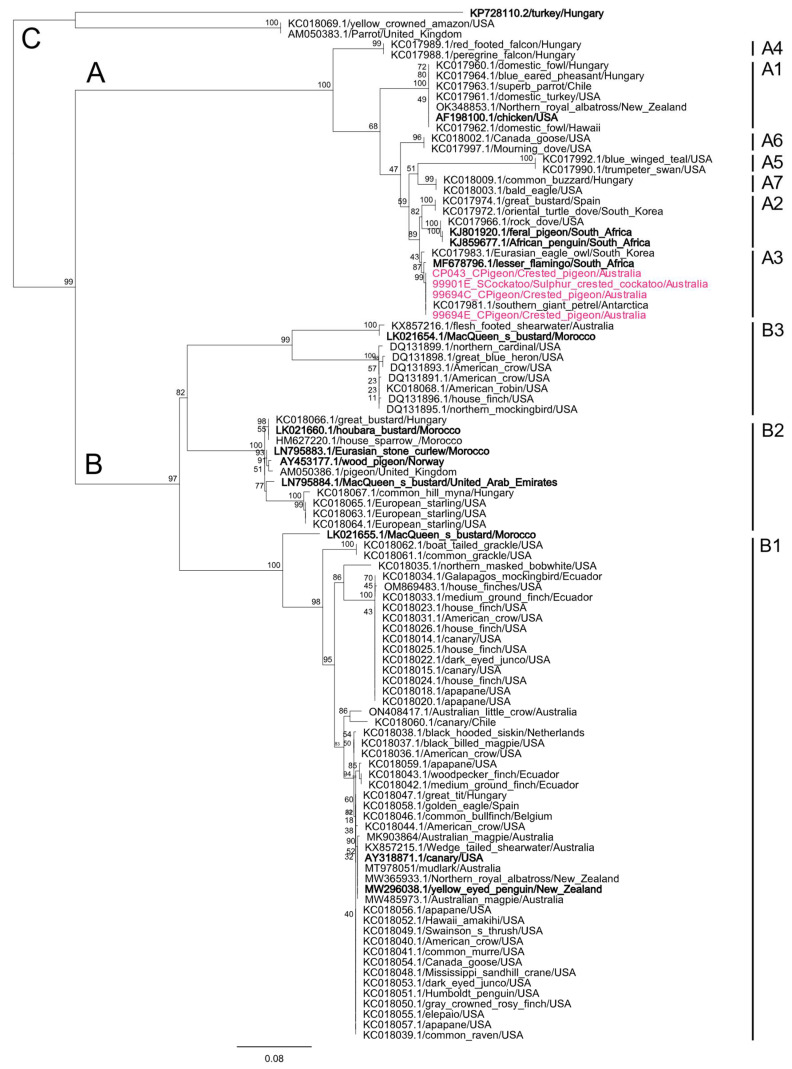
Maximum likelihood (ML) phylogenetic tree from partial nucleotide sequences of the P4b gene from four *Avipoxvirus* sequences detected from avian species in this study and global *Avipoxvirus* reference strains. ICTV-ratified species are indicated in bold black text. Major clades and subclades are designated according to Gyuranecz et al. [28], with branches labelled (**A**–**C**) indicating major *Avipoxvirus* clades and subclade names are provided adjacent to tip names. Branch tip labels show the GenBank accession number/common name/country of origin, with *Avipoxvirus* sequences detected from this study being boldened in pink. Bootstrap values are shown as a percentage. The scale bar indicates the number of nucleotide substitutions per site.

**Figure 6 viruses-15-00143-f006:**
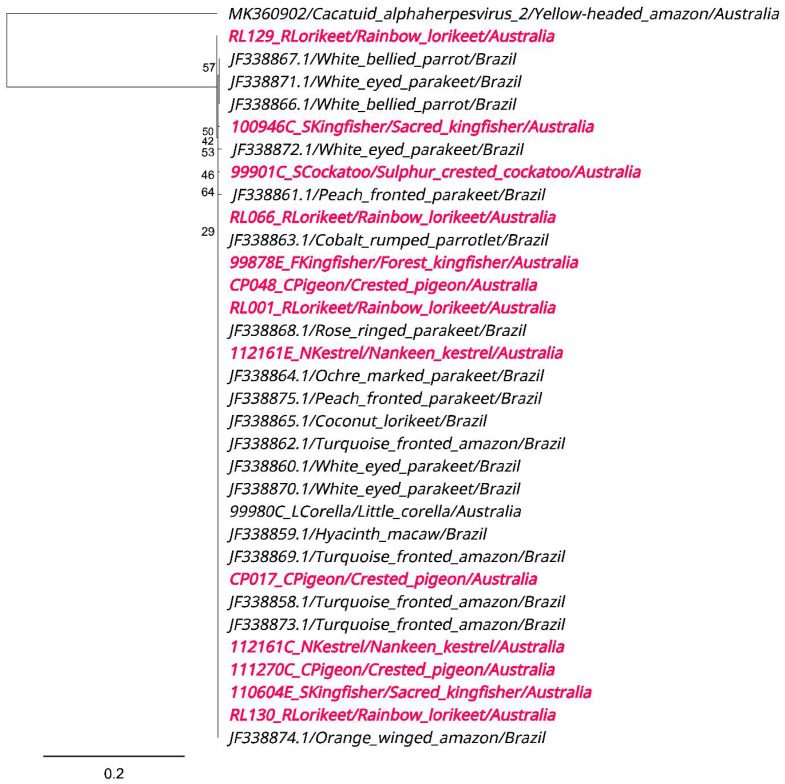
Phylogenetic relationships between 14 PsAHV1 UL16/17 gene sequences detected from various avian species in this study and 19 Brazilian reference sequences. The ML phylogenetic tree was constructed using IQTREE2, using *cacatuid alphaherpesvirus 2* strain 97-0001 (MK360902) as the outgroup. Branch tip labels show the GenBank accession number/common name/country of origin. PsAHV1 sequences detected from this study are also coloured in pink and boldened, while reference sequences remain in regular font and are coloured black. The scale bar indicates the number of nucleotide substitutions per site. Bootstrap values are shown as a percentage on the left.

## Data Availability

The *Avipoxvirus*, BFDV and PsAHV1 sequences from this study were deposited in GenBank: OP131423–OP131511 (BFDV), OP13512–OP131515 (*Avipoxvirus*), OP146304–OP14317 (PsAHV1).

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
