# Peer review of "Unexpected Pathogen Diversity Detected in Australian Avifauna Highlights Potential Biosecurity Challenges"

_viruses, 2023, doi:10.3390/v15010143_

Round 1
Reviewer 1 Report
Dear Authors
Manuscript can be accepted
Minor revision

Author Response
We thank the reviewer for taking the time to read the manuscript and for providing constructive comments. Please see attached word document with detailed responses.

Reviewer 2 Report
Kasimov et al. present original research on pathogen diversity in birds in Australia. Overall, I am confident in the PCR and sequencing work and in the descriptive data presented. This information helps wildlife biologists and conservation biologists consider where risks may be, and where they may not be, in pathogens among bird populations. My only concern is the amount of data presented here that was already published elsewhere.
Abstract
Lines 21-24 The choice to italicize some virus names and not others is confusing. I do not think that beak and feather disease virus are supposed to be italicized, nor the two alphaherpesviruses?
Line 72 the birds are described as “wild birds’, but were they not collected from a wildlife rehabilitation facility? This needs to be clear throughout due to potential bias from sampling only in that group of birds.
Line 95 I recommend rewording this, perhaps each bird was assigned to a taxonomic group or the taxonomic group was recorded. Each individual was not divided.
Figure 1a. I am not sure how useful the Gini-Simpson index score is here. This is almost certainly not all of the taxonomic groups that were admitted to the AZWH, it was a select group, therefore, the sampling protocols that are useful in pairing with a Gini-Simpson index are not in play here.
Lines 129-131 The inclusion of so much data from a previous study is potentially a double-publishing issue. The authors should cite the other study, but these data should stand alone. If there is not sufficient data for it to stand alone, perhaps it should not be published. This seems to apply mostly (perhaps only) to the Chlamydiaceae, and it is clear it further complicates the work with the methods notes in Lines 173-174.
Results/Section 3.1
I think this all belongs in the Methods section. These are parameters of the study, not results.
Author Response
We thank the reviewer for taking the time to read the manuscript and for providing constructive comments. Please see the attached word document with detailed responses.
